

# A new dataset of Southern Ocean sea-ice leads: First insights into regional lead patterns, seasonality and trends, 2003–2023

Umesh Dubey, Sascha Willmes, and Günther Heinemann

Department of Environmental Meteorology, Trier University, Trier, Germany

**Correspondence:** Umesh Dubey (dubey@uni-trier.de) and Sascha Willmes (willmes@uni-trier.de)

**Abstract.** Sea-ice leads play a key role in the climate system by facilitating heat and moisture exchanges between the ocean and atmosphere, as well as by providing essential habitats for marine life. This study presents a new dataset on monthly sea-ice leads in the Southern Ocean and a first comprehensive analysis of spatial patterns, seasonal variability, and long-term trends of wintertime (April to September) sea-ice leads over a 21-year period (2003–2023). Our findings reveal that leads are

ubiquitous in the Southern Ocean and show distinct spatial patterns with maximum lead frequencies close to the coastline, over the shelf break and close to seafloor ridges and peaks. We see a strong seasonal variability in lead occurrence, with lead frequencies peaking in mid-winter. Weak but significant trends in lead frequencies can be inferred for the presented period for individual regions and months. Rather small changes in lead occurrence over the 21 years suggest rather stable wintertime sea-ice compactness despite the observed strong fluctuations and recent anomalies in sea-ice extent. This study provides first

results on the spatial and temporal dynamics of sea-ice leads in the Southern Ocean and can thereby contribute to an improved understanding of air-sea ice-ocean interactions in the climate system. It also underscores the need for further investigation into the individual contributions of atmospheric and oceanic drivers to sea-ice lead formation in the Antarctic.

## 1   Introduction

The Southern Ocean is a key component of the climate system, and its sea-ice cover is characterized by a dynamic pattern of

leads – narrow, elongated openings in the pack ice. Understanding the temporal variability and spatial distribution of leads is essential due to their role in facilitating significant exchanges of heat and moisture between relatively the warm ocean and cold winter atmosphere (Alam and Curry, 1995; Marcq and Weiss, 2012; Lüpkes et al., 2008; Heinemann et al., 2022). In addition to their impact on ocean/sea-ice/atmosphere interactions, leads are valuable diagnostic parameters for sea ice drift (Spreen et al., 2017; Kwok et al., 2013) and provide habitats for marine mammals and birds (Stirling, 1997). Moreover, increasing sea ice

deformation and lead opening can accelerate sea ice thinning through the sea ice-albedo feedback (Curry et al., 1995; Eicken and Lemke, 2001). Ice-free areas including sea-ice leads have also been identified as sources of methane emissions (Kort et al., 2012; Damm et al., 2015).

Leads are not uniformly distributed in the sea ice. Patterns in their spatial distribution can be influenced by atmospheric forcing in the short-term, such as wind speed and wind divergence, and by ocean processes in the long-term (Willmes et al.,

2023). These leads, forming and evolving due to complex interactions between atmospheric and oceanic processes, represent





an important component for understanding climatic variability and trends in polar regions (Wang et al., 2016; Zhang, 2014; Rheinlaender et al., 2022). Hence, studying lead dynamics is relevant for gaining better insight into the feedback mechanisms within the Southern Ocean climate system.

While distinct patterns in the spatial distribution of leads have recently been identified in both polar regions (Reiser et al., 2019; Willmes et al., 2023), the spatial and temporal variability and trends in wintertime sea-ice lead dynamics for individual regions in the Southern Ocean are yet to be determined. The potential of thermal-infrared satellite data for the detection of sea-ice leads has been demonstrated for the Southern Ocean in a previous study (Reiser et al., 2020). The applied lead detection works well during the winter months when there is a high temperature contrast between the warm ocean and a cold atmosphere. As a result, leads can be detected on a daily basis at a spatial resolution of 1 km$^2$, however, with gaps due to clouds.

The primary objectives of this study are to deduct a continuous and gap-free monthly lead dataset from the daily lead data obtained from Reiser et al. (2020) and to use this dataset to infer the spatial distribution of sea-ice leads, quantify temporal trends and compare regional differences for the 21-year period from 2003 to 2023. By addressing these objectives, this research contributes to a deeper understanding of the mechanisms driving sea-ice dynamics and associated climatic implications. By enhancing our understanding of sea-ice leads in the Southern Ocean and their relationship with atmospheric and oceanic forcing, this work aims to contribute a more comprehensive picture of air/sea-ice/ocean interactions. The presented data can provide valuable contributions for improving sea ice/ocean components in climate models and forecasting systems, which are essential for predicting future changes in polar environments and their global implications.

The structure of the study is as follows: The data and methods used to derive the monthly lead dataset and to conduct a preliminary analysis are described in Section 2. Following this in Section 3, we identify regions with the highest frequency of leads and show the predominant spatial and temporal lead patterns in wintertime sea ice. We also examine trends and anomalies across various regions in the Southern Ocean and discuss our results in Section 4. Finally, we conclude our work in Section 5.

## 2   Data and Methods

### 2.1   Sea-ice lead data

The detection of sea-ice leads from thermal-infrared satellite data is based on identifying significant positive surface temperature anomalies. This method, as described by Reiser et al. (2020), classifies each pixel into one out of four classes: sea ice, clouds, artefacts and leads. We use these daily lead maps on a 1 km$^2$ resolution grid to calculate monthly lead frequencies (LF) in the months of April to September for the years of 2003 to 2023. The monthly aggregated maps are the main database for the presented analysis in this study. Daily sea ice concentration from passive-microwave data from the University of Bremen (Spreen et al., 2008) is used to define a mask by applying a sea ice concentration threshold of 15 %, which is a common value to determine the sea-ice extent. It is important to note that our dataset focuses on small-scale leads and does not capture large-scale polynyas. Areas being classified as open water in the passive microwave sea-ice concentration data will not be detected as leads in our dataset. For studies including polynya areas, we suggest to combine different datasets.




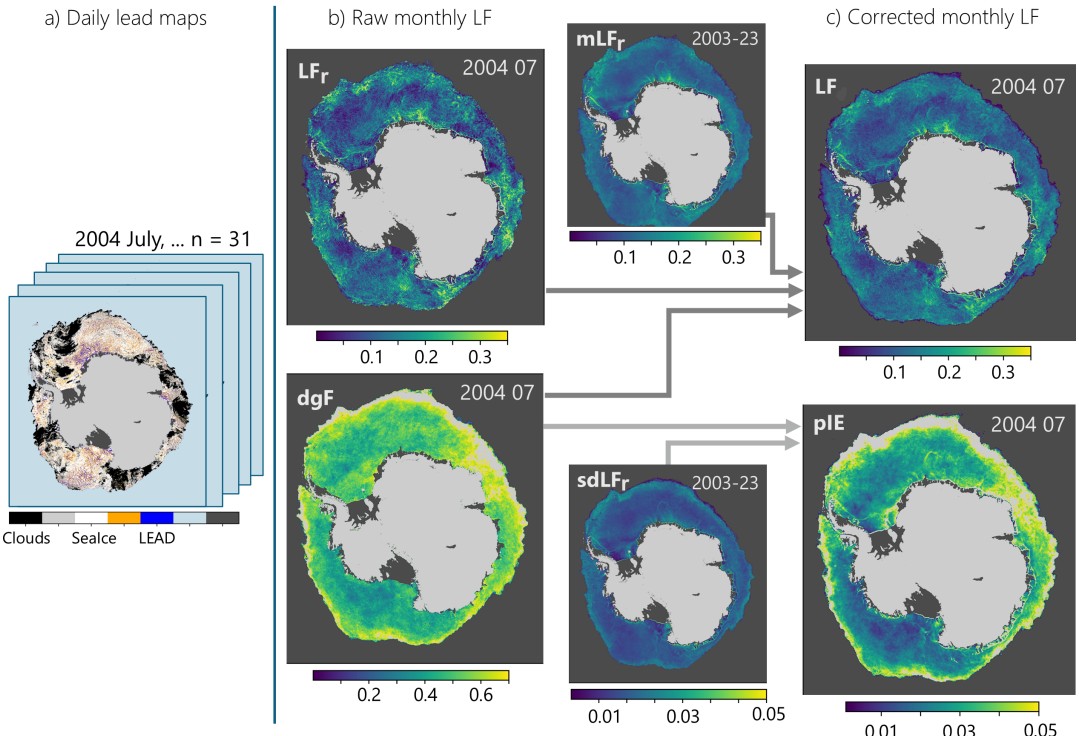

**Figure 1.** Retrieval of monthly lead frequency (LF) from daily lead maps: **(a)** Daily lead maps for July 2004 with categories cloud, sea ice, lead, artefact, **(b)** raw monthly LF ($LF_r$) and data gap frequency (dgF) derived from daily data, and **(c)** corrected monthly LF and the potential Integration Error (pIE) derived from the overall mean $LF_r$ ($mLF_r$), the standard deviation of $LF_r$ ($sdLF_r$), and monthly dgF.

## 2.2 Retrieval of monthly lead frequency

The LF is a temporally integrated quantity that represents the number of days a pixel is covered by a lead during a specified
period, relative to the number of available clear-sky (sea ice or lead) observations (Willmes and Heinemann, 2016). This aggregation provides monthly, annual and total LF values for the Southern Ocean. The retrieval of monthly LF involves several steps, starting with the daily lead maps, as shown in panel (a) of Fig. 1. As the daily lead maps are derived from satellite thermal-infrared data, valid lead detections are only available for clear-sky conditions. This means that cloud cover introduces gaps in the data, and the uncertainty in monthly LF calculations increases with the number of cloud gaps in the monthly stack.
An example from July 2004 is shown here to demonstrate how daily data are influenced by cloud cover and the resulting data gaps.

In Fig. 1b, the raw monthly LF ($LF_r$) is calculated by integrating daily lead maps over a month. The number of data gaps in the monthly stack, represented by the data gap frequency (dgF), influences the confidence of the applied integration. Potential biases in $LF_r$ calculations due to high dgF values need to be taken into consideration. This is addressed by applying an LF



correction scheme together with an uncertainty measure for the monthly LF values. Therefore, we first derive basic statistical quantities of monthly $LF_r$, i.e.: the long-term mean ($mLF_r$) and standard deviation ($sdLF_r$). Then a correction method is applied to obtain the monthly corrected LF (see Fig. 1c), which essentially represents a weighted smoothing of LF values based on the number of data gaps (dgF) per pixel (see Eqs. 1 and 2).

$$LF = LF_r - dgF \cdot (LF_r - mLF_r) \pm pIE \qquad (1)$$

$$pIE = dgF \cdot sdLF_r \qquad (2)$$

By this, we assume large $LF_r$ deviations from the long-term mean to be potentially biased by data gaps and the associated error. The correction offset becomes larger with an increasing $LF_r$ anomaly and higher dgF values and always pushes LF towards the climatological mean. This approach aims to avoid an over-interpretation, but to increase confidence for the observed signals, i.e. LF anomalies, instead.

We also define the absolute potential Integration Error (pIE) to measure the associated uncertainty in LF due to a combination of dgF and the local standard deviation of $LF_r$ ($sdLF_r$) from the monthly climatology. A larger local $sdLF_r$ and a higher dgF will thereby increase the uncertainty. The pIE represents a potential absolute LF error and will here be used to constrain an analysis of spatial-temporal LF patterns to pixels and months when the observed LF anomaly (LFA) was larger than the associated pIE (hereafter referred to as the LFA > pIE criterion, with LFA < pIE indicating low-quality LF data). It thereby

serves as a metric to exclude low-quality LF data in quantitative investigations (e.g., section 3.2). The obtained monthly LF anomalies are independent from anomalies in cloud cover and the associated data gaps (see Appendix, Fig. A1), which is important to make sure that the obtained LF variability is not a function of available clear-sky data.

### 2.3  Auxilliary data

For a preliminary evaluation of confidence in the identified lead patterns, we use ocean surface current data, provided by the

European Centre for Medium-Range Weather Forecasts (ECMWF) Ocean Reanalysis System 5 (ORAS5) dataset (Zuo et al., 2019). ORAS5 provides global ocean reanalysis products with monthly surface current velocity data at a spatial resolution of 0.25°. These data include estimates of zonal and meridional current velocities, offering insights into large-scale ocean current velocity patterns. For this study, we use ORAS5 surface current data with the climatological mean of LF to calculate the maximum coincident percentile exceedance (CPE) as a measure of spatial similarity. CPE can be used to identify areas where

ocean currents modulate lead formation and variability.

The bathymetry data used in this study is derived from the International Bathymetric Chart of the Southern Ocean (IBCSO) version 2.0 (Dorschel et al., 2022), a digital bathymetric model (DBM) for the area south of 50° S in the Southern Ocean. It has a horizontal resolution of 500 m. Similar to the use of ORAS5 data, we use bathymetric gradients and mean LF to compute CPE and thereby highlight regions where bathymetry is obviously related to the formation of leads. The results of this comparison

are shown in section 3.8.



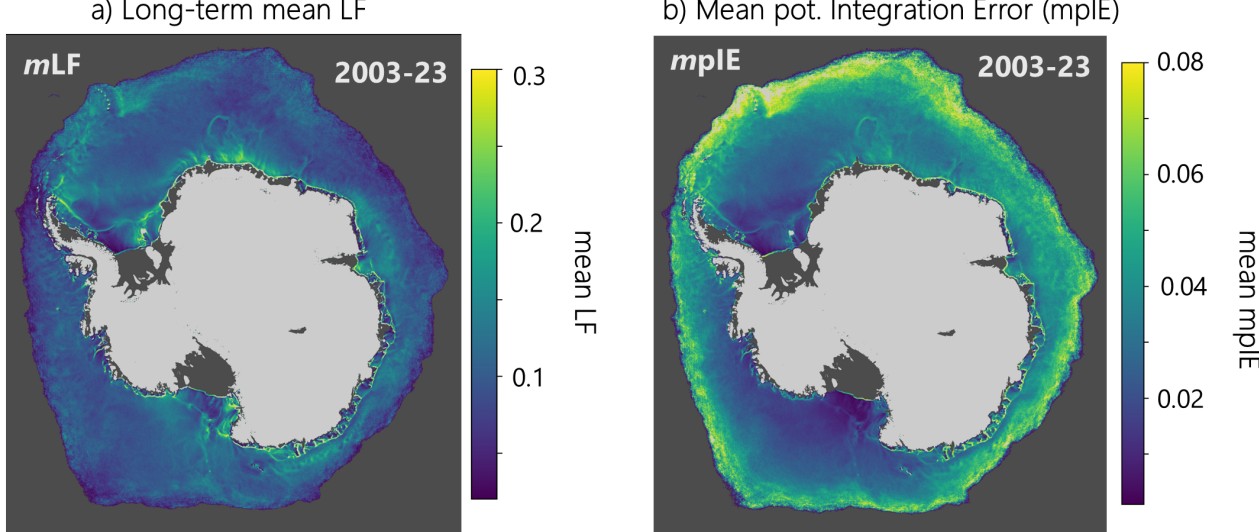

**Figure 2. (a)** Overall mean of corrected LF (mLF) and **(b)** overall mean absolute potential Integration Error (mpIE) for the months of April to September in the period from 2003 to 2023.

## 3 Results

### 3.1 Spatial patterns of sea-ice leads in the Southern Ocean

Figure 2 shows the long-term mean of the corrected LF (mean LF) over the 2003–2023 period, which essentially represents an updated and improved version of the lead climatology shown and discussed in Reiser et al. (2019). This visualization highlights the spatial patterns in lead frequency across the Southern Ocean. The LF value indicates how often on average a lead was present during wintertime at a specific position in the 21-year period.

The most dominant spatial lead patterns are mainly found in Weddell and Ross Seas along the shelf break with values exceeding 0.25, which means that leads were present on more than 25 % of days during winter. Lead hotspots often follow linear patterns, with a prominent line of increased LF running close to the Antarctic coastline. In regions such as the Weddell Sea, Prydz Bay and Ross Sea, further areas of concentrated lead activity are found along the continental shelf break and deep sea ridges, such as Maud Rise and Gunnerus Ridge. Close to Maud Rise we see what is generally referred to as warm-water "Halo" around Maud Rise and represents a ring-like structure of high LF values at the location where the Maud Rise polynya is frequently forming (note that larger polynyas are not included in this dataset). The presence of bathymetric features, such as ridges and troughs, align with these regions of increased LF, which confirms the findings by Reiser et al. (2019). This agreement suggests that the seafloor topography plays a crucial role in the formation and persistence of leads. The distinct band of increased LF at the coastline and at the edge of fast-ice regions likely reflects the interaction between ocean currents, wind patterns and ice dynamics, particularly in areas where the ice is thinner or more mobile (Wang et al., 2023). The associated





overall mean potential Integration Error (mpIE) in Fig. 2b identifies areas that are more prone to data gaps and potential errors in the monthly LF integration. The mpIE apparently becomes generally larger towards the marginal ice zone and in areas with

increased sdLF$_r$. It represents the potential error in the integrated monthly LF and does not exceed values of 0.07 in most regions of the inner sea ice pack (south of the Marginal Ice Zone, MIZ).

## 3.2    Inter-annual variability of leads

Figure 3 shows annual wintertime LF anomalies in the Southern Ocean from 2003 to 2023, with respect to the overall mean LF, from April to September, 2003–2023. Areas, where the LFA > pIE criterion is not met, are highlighted in green color

(low-quality). In all of the presented years, a mixture of positive and negative anomalies is found with their magnitude and regional differentiation changing from year to year. Also the total area of low-quality data (LFA < pIE, green color) is subject to inter-annual changes. The years of 2021–2023 exhibit notably larger low-quality LF areas as compared to the previous years. These years are obviously affected by more data gaps in a broader region south of the MIZ. No fixed pattern in LF anomalies can be found. It is possible, however, to identify some years with remarkable regional lead anomalies. For example, in 2006

there were substantially more leads in the Weddell Sea than in the previous and following years with anomaly values partly exceeding 0.05, which means that leads were found here during roughly 10 more days of the 182-days period. Lead occurrence in the Ross and Amundsen Seas was especially strong in 2004, 2012, 2013 and 2016. LF anomalies after 2020 seem to be generally larger than in previous years. This comes, however, with increased pIE and associated low-quality LF areas. There is no year with an overall positive or negative anomaly, but both occur in different regions of the Southern Ocean in each of the

presented years.

## 3.3    Seasonal variability of leads

The seasonal evolution of the monthly mean LF anomalies is given in Fig. 4. These anomalies provide insight into the general seasonal variability of sea-ice leads in the Southern Ocean. In April, when new sea ice starts to consolidate, negative and close-to-zero anomalies indicate that leads are less abundant as compared to the total winter mean in most of the regions. Only

in the southern Weddell Sea and in coastal regions leads seem to be more frequent in April on average. From May to June, leads become more frequent also in the Ross and Bellingshausen-Amundsen Seas. In June, the maximum occurrence of leads south of the MIZ is observed, i.e.: leads are more abundant in June than in any other month of the winter season. In mid-winter, July and August, sea ice seems to consolidate, and leads are forming predominantly in the marginal ice zones, while in August LF is found to increase in the southwestern Ross Sea on average. In September, the overall lead activity seems to cease.

However, in contrast, the coastal regions in the East Antarctic, with a pronounced positive anomaly north of the Dronning Maud and Enderby land are characterized by an increased LF during this time of the year. The fraction of areas covered by low-quality data (LFA < pIE) is increasing throughout the season. In August, almost the entire Eastern Antarctic sea-ice region is affected, except the coastal regions. In September, confident anomalies are limited to the southern Weddell Sea (negative) and the pronounced positive anomaly north of Dronning Maud and Enderby land mentioned above. Overall, the seasonal cycle





**Figure 3.** Annual lead frequency anomalies of wintertime LF with respect to the 21-year (2003–2023) climatological mean. Areas, where the LFA > pIE criterion is not met, are highlighted in green color.

of lead occurrence in the Southern Ocean can be summarized as increasing from April to June and then decreasing again until September.



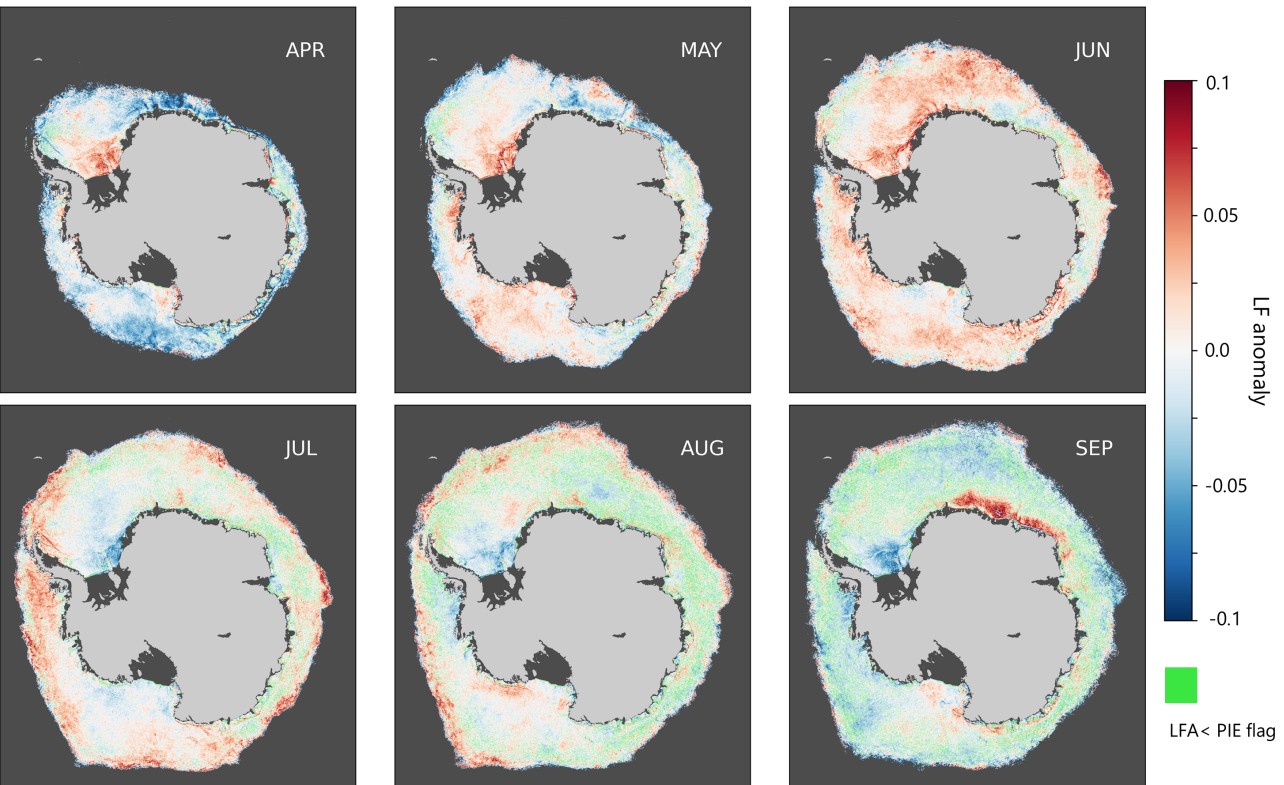

**Figure 4.** Seasonal variability of monthly mean lead frequency anomalies from April to September, 2003–2023. These anomalies are calculated relative to the 21-year climatological mean. Areas, where the LFA > pIE criterion is not met, are highlighted in green color.

## 3.4 Seasonal and sectoral variability of leads

The time series plots in Fig. 5 illustrate the inter-annual variability of monthly LF in the Southern Ocean and its sub-regions, i.e.: the Weddell Sea, Ross Sea, Indian Ocean, Pacific Ocean and Bellingshausen-Amundsen Seas from 2003 to 2023. Each
plot represents monthly mean LF values (distinguished by different colors) for April to September. Data points without color fill in the markers indicate months, where the valid LF data coverage (i.e., LFA > pIE) for the LF mean in the specific region, is lower than 50 %, which means that the data-quality is low.

A strong seasonal variability is confirmed here with low monthly mean LF in April (blue) and high LF in June (red). The LF variability across all sectors suggests that significant anomalies occur primarily on a monthly basis. The time series reveals
mostly increasing LF values over the 21-year period, mainly in the Southern Ocean, Weddell Sea and Indian Ocean. It has to be noted, however, that a large part of this observed increase is attributed to an increase in low-quality data points during the last 3 years of the observed period. While low-quality is not necessarily the cause for increased LF (low-quality data points are also found in months with low LF, see also Fig. A1), it still reduces confidence in a retrieval of linear trends from the presented time





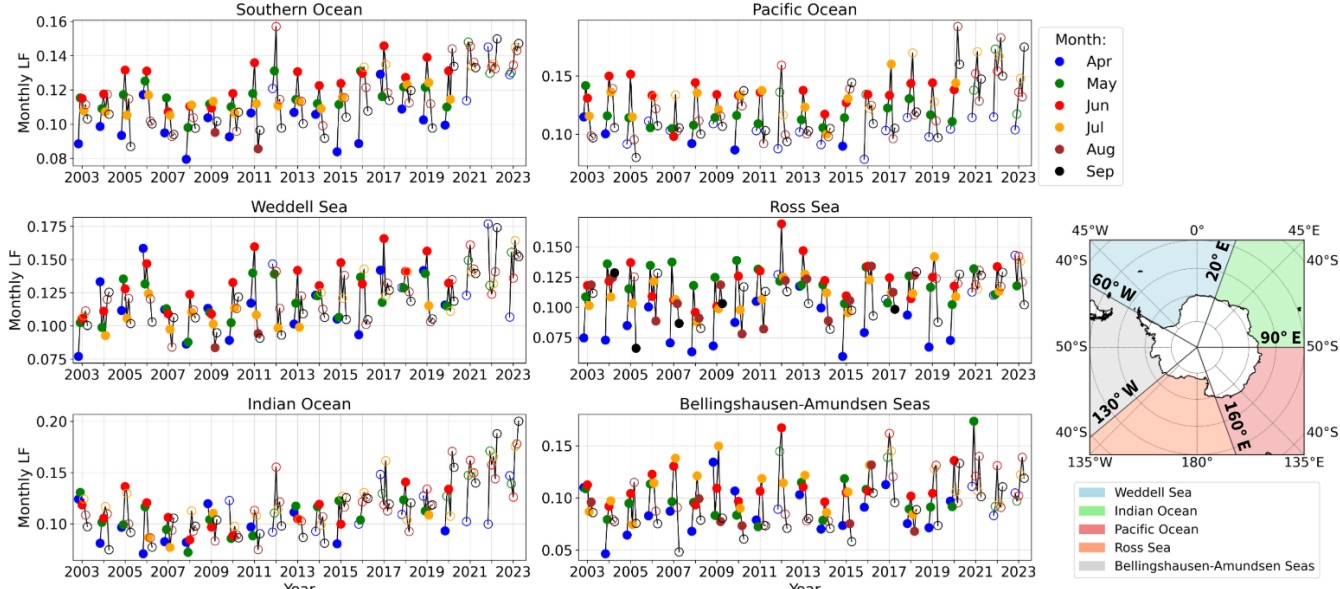

**Figure 5.** Inter-annual variability of monthly mean lead frequency (April to September) in the Southern Ocean and its sub-regions from 2003 to 2023. Sub-regions are identified in the inset map. Individual months are highlighted by color. Month dots without color fill indicate low-quality (LFA < pIE) data point coverage above 50 %. Note the different scales on the y-axes for monthly LF.

series. In the above mentioned sectors, a weak but obvious LF increase can be determined also when the low-quality data points
are not taken into account (for a detailed trend analysis see section 3.6). Some periods stand out by pronounced deviations of the monthly mean LF values. Most obvious is the two-year period of 2012 and 2013 with substantially higher monthly mean LF in the Ross Sea in all months. Also, June 2012 and May 2012 stand out with high LF in the Bellingshausen-Amundsen Seas. In the Indian and Pacific Ocean sectors, low-quality data points are more frequent, which is a result of their rather narrow area and proximity to the MIZ with increased data gaps (high pIE).

**3.5 Temporal evolution of lead frequency anomalies**

The regional and temporal lead variability is given in more detail in latitude-averaged monthly LF anomalies, categorized into 5° longitude bins, covering the period from April 2003 to September 2023 as presented in Fig. 6. The analysis of meridional and zonal lead patterns between 60° S and 90° S as shown in Fig. 6a provides a comprehensive view on how lead dynamics vary across longitudes in the Southern Ocean over time (the coastlines below Fig. 6a represents the corresponding positions
along the longitudes). The most notable aspect is the obvious presence of "strong lead events" and "low lead events", which are highlighted by pronounced positive anomalies (red areas) and negative anomalies (blue areas), respectively. Four of these events have been selected and are illustrated as monthly LF anomaly maps in sub-figures 5b–e. Low-quality data (green color) are mostly present in the Pacific and Indian Sectors (0° E – 135° E) and increase substantially after 2020, accompanied by an



**Figure 6. (a)** Latitude-averaged (60° to 90° S) monthly lead frequency anomalies in 5° longitude bins from April 2003 to September 2023. Areas, where the LFA > pIE criterion is not met, are highlighted in green color. **(b–e)** Monthly LF anomaly maps of selected years and months from the left panel **(a)** are shown. Strong lead anomalies can be seen, for example, in **(b)** April 2006, **(d)** June 2012 and **(e)** May 2021. Anomalies are calculated relative to the climatological monthly mean.

overall increase of LF. Some examples for the pronounced regional anomalies are depicted by arrows. The map for April 2006
(Fig. 6b) shows a significant positive anomaly, with values reaching up to 0.1, over the Weddell Sea at around 45° W. In June
2008 (Fig. 6c), a negative LF anomaly is observed along the coastal area in the Cosmonaut Sea near 50° E longitude in the



Indian Ocean. The map for June 2012 (Fig. 6d) reveals a notable positive LF anomaly of more than 0.1 along the Dronning Maud coast of the East Antarctic, particularly between 0° to 45° E longitude. A positive LF anomaly of less than 0.12 is evident in the Bellingshausen-Amundsen Seas during May 2021 (Fig. 6e). These maps just exemplarily show the spatial patterns of

lead anomalies on the monthly scale. As Fig 6a reveals, more of these events can be identified for different regions throughout the 21-year period.

The variability of lead dynamics shown here indicates how monthly anomalies are superimposed to the previously described general seasonal, inter-annual and long-term spatial patterns. Regional monthly anomalies provide a good basis to identify atmospheric drivers for sea-ice break-up and lead dynamics (see section 4: Discussion).

### 3.6  Trend analysis of leads

Table 1 presents the overall change in monthly mean LF in the Southern Ocean and its sub-regions from 2003 to 2023. The summarized LF changes are based on significant linear trends at the 5 % level ($p$ values < 0.05). Each cell in the table shows the change in LF for the corresponding region and month within the 21 years covered by the used dataset. The given LF change is the mean change per region and month for all the pixels, in which the slope value is significant. To avoid an over-interpretation

of trends, the low-quality data points (compare Fig. 5) where replaced by the overall monthly mean before trends were derived. As a result, we obtain LF changes in the period from 2003–2023 as summarized in Table 1. The changes are significant, but small across the Southern Ocean and its sub-regions. When the entire area is considered (Southern Ocean), the mean change is positive during all months with a maximum in July (+0.05). The maximum LF changes are found in the Pacific Sector in September (+0.1) and April (+0.08). The Bellingshausen-Amundsen Seas exhibit modest increases, especially in August and

September (0.07 each). Negative changes are found less frequently, with the strongest LF change observed in the Ross Sea in May (–0.07). In the Western Antarctic, a decrease in LF is only found in the Weddell Sea for April (–0.05).

**Table 1.** Change in monthly mean LF in the period from 2003 to 2023 per region and month for pixels showing significant ($p$ values < 0.05) slopes in linear trends. Each value represents the overall change within 21 years according to the derived slope of a linear trend.

| Month | Southern Ocean | Weddell Sea | Indian Ocean | Pacific Ocean | Ross Sea | Bellingshausen-Amundsen Seas |
|---|---|---|---|---|---|---|
| APR | 0.02 | –0.05 | 0.02 | 0.08 | 0.06 | 0.01 |
| MAY | – | 0.01 | 0.05 | 0.04 | –0.07 | 0.03 |
| JUN | 0.02 | 0.03 | 0.04 | –0.01 | 0.01 | 0.01 |
| JUL | 0.05 | 0.05 | 0.02 | 0.06 | 0.05 | –0.01 |
| AUG | 0.04 | 0.04 | 0.05 | 0.04 | – | 0.07 |
| SEP | 0.03 | 0.03 | 0.03 | 0.10 | – | 0.07 |



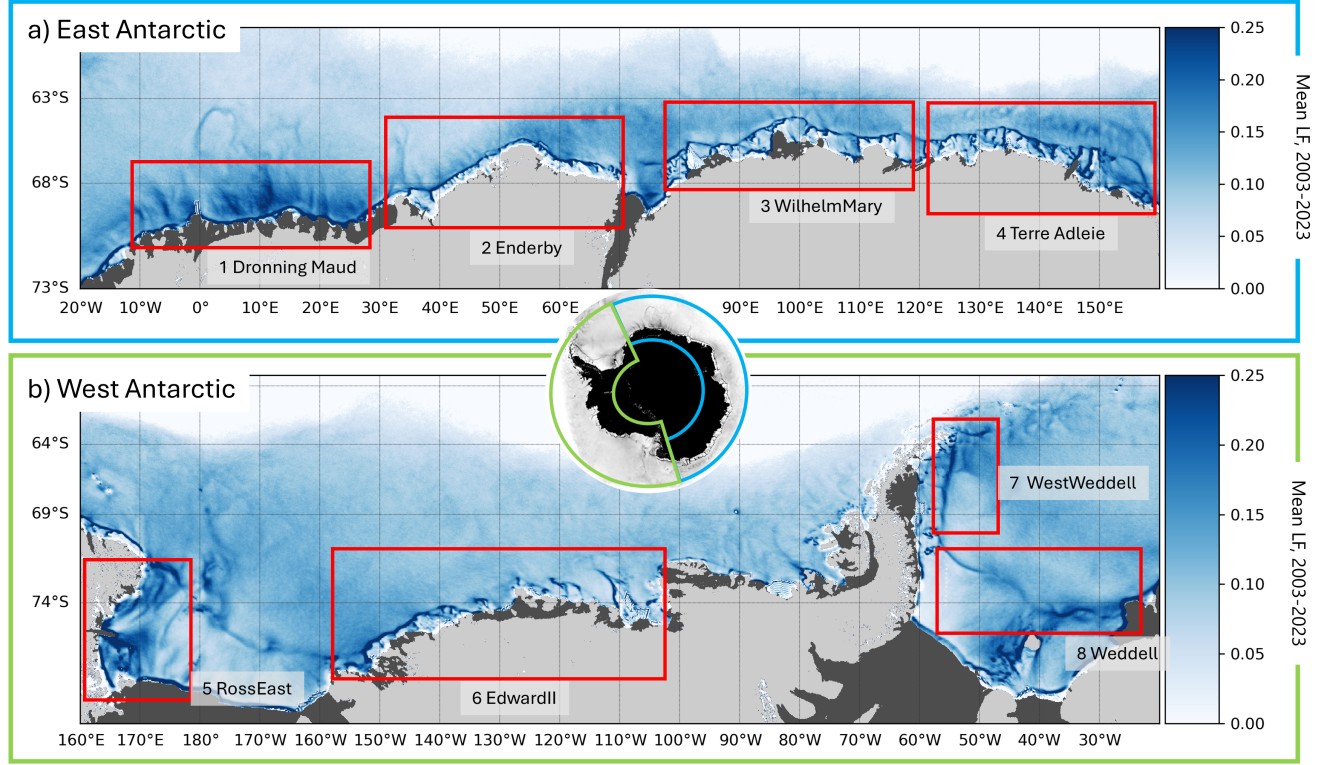

**Figure 7.** Mean LF for wintertime period 2003–2023 in coastal regions for **(a)** the East Antarctic and **(b)** the West Antarctic. Red boxes indicate sub-regions used for the regional analysis of individual inter-annual and seasonal lead variabilities in Figs. 8 and 9.

## 3.7 Leads in coastal areas

As leads are especially abundant in coastal regions and at fast-ice edges (flaw-leads), we also want to focus on these areas to highlight coastal region lead dynamics based on the presented dataset. Figure 7 illustrates the spatial patterns of mean LF

across various regions of the East and West Antarctic during the period from 2003 to 2023. Box 8 is an exception as it is not a coastal region but lies over the pronounced band of high LF over the Weddell Sea shelf break. The spatial LF patterns clearly reveal several sea-ice features that are typical for coastal regions. Particularly noticeable is the pronounced band of flaw-leads that is found right in front of the ice shelves, but also at the edge of fast-ice areas, which become visible here through low LF values in the used dataset (see for example the red boxes 2, 3 and 4 In Fig. 7a). The LF values here can be interpreted as inverted

fast-ice frequencies, with low LF representing high fast-ice occurrence and vice versa. Fast-ice, flaw-leads and polynyas can also be recognized in the West Antarctic (Fig. 7b). Overall, the spatial patterns across the East and West Antarctic highlight pronounced regional differences in lead occurrence and the presence of fast-ice areas. Coastal polynyas, which are regions of persistent open water or thin ice, are mainly driven dynamically by wind systems, such as cold winds from the ice shelf or ice sheet (Paul et al., 2015), causing high heat energy fluxes compared to the surrounding pack ice (Marcq and Weiss, 2012). The





**Figure 8.** Mean monthly LF per month and year (left) with open dots indicating months with an LFA < pIE pixel coverage above 50 %, times series of monthly LF with blue open dots indicating months with an LFA < pIE pixel coverage > 50 % (middle) and mean seasonal LF change (right) with its standard deviations (vertical lines) for each month from April to September, 2003–2023. Sub-figures a–d indicate sub-regions of the East Antarctic (red boxes) shown in Fig. 7a.

continuous exchange of heat and moisture in these regions further influences local sea-ice dynamics, potentially maintaining higher lead frequencies. As a result, coastal areas are generally exhibiting higher LF values than the inner ice-pack, the shelf breaks and some other bathymetry hot spots (e.g., Maud Rise Halo) excluded. The observational data summarized in Fig. 7 are useful to expand upon observations of long-term polynya areas for example in Lin et al. (2024).

Figures 8 and 9 show the seasonal and inter-annual variability of mean monthly LF in the sub-regions indicated by red boxes in Fig. 7 during the 2003–2023 winter seasons. The heat maps (left) highlight regional monthly changes in LF and indicate







**Figure 9.** Same as in Fig. 8 but for sub-regions of the West Antarctic shown in Fig. 7b.

some pronounced lead events, e.g., June 2012 and 2013 in the Droning Maud region, low LF at the coast of Enderby land in April between 2005 and 2007, high LF in July 2018 at Terre Adelie land, and generally an overall increase in LF during September after 2020 in most of the regions. As in Fig. 5, this obvious increase is accompanied by a stronger presence of low-quality (LFA < pIE) data. The dots in the presented heat maps are representative of months, in which the regional coverage of these low-quality LF data exceeded 50 % of the box area. The time series plots (middle) indicate the inter-annual LF variability for each sub-region and highlight again the pronounced increase in LF after 2020. This increase was only found in the Eastern Weddell Sea (boxes 1–4), while in the West Antarctic, no significant changes are notable in the 21-year period. As strong presence of low-quality data points (LFA < pIE above 50 %) is here indicated by dots without color fill. The mean seasonal changes in LF per region (right) offer a detailed overview on regional differences in the seasonality of lead dynamics. While




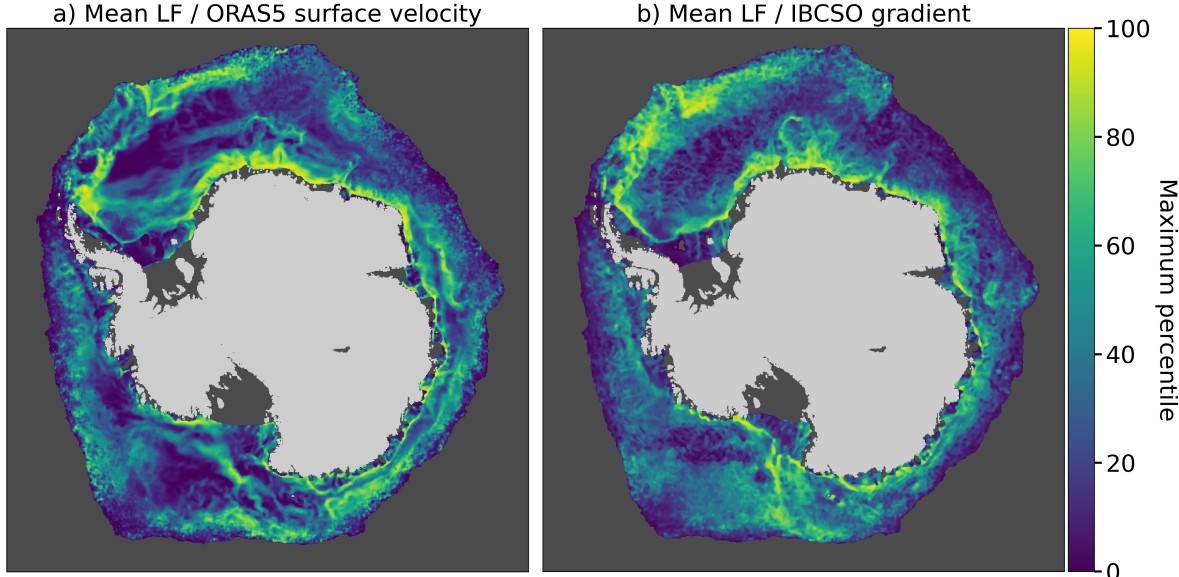

**Figure 10.** The maximum coincident percentile exceedance (CPE) for overall mean LF and **(a)** ORAS5 surface current velocity (m/s) and **(b)** IBCSO2 bathymetry gradient (m). Values indicate the maximum percentile per pixel that is exceeded coincidentally in both datasets.

mean LF increases in boxes 1 and 2 (Dronning Maud and Enderby) from April to September, it shows differing evolutions in the other presented regions. Box 8 (WeddellSlope) stands out here with a strong drop in average LF in July and low values also in August and September.

### 3.8 Spatial lead patterns in context with surface currents and ocean depth

Here, we aim to evaluate the spatial similarity in local maxima of the climatological mean LF (see Fig. 2a) to ocean surface currents and bathymetry gradients. We therefore use the CPE (see Section 2.3) as a metric of comparison, which represents the common maximum percentile that is exceeded in both datasets at a given grid point. High values in CPE indicate areas where both datasets show high values coincidentally.

Figure 10a illustrates the spatial patterns of CPE between mean LF and mean ORAS5 surface current velocity in wintertime 2003–2023. The map highlights regions where CPE values exceed high percentiles in both datasets. High CPE values (of more than 80 %) are observed mainly in regions of the Antarctic Slope Current (ASC) and close to the MIZ, particularly along the Weddell Sea, Ross Sea, and Indian Ocean sectors. The highest values are concentrated near the Antarctic coastal zones, where the mean surface currents and LF are most pronounced. The CPE map also highlights regions close to seafloor ridges and peaks (Maud Rise, Gunnerus Ridge, etc.), which are both characterized by high LF and strong surface currents at the same time. This is demonstrated even better in Fig. 10b, which shows the relationship between mean LF and bathymetric gradients derived from IBCSO version 2.0 data. High CPE values are predominantly observed in areas where significant changes in ocean depth occur,





such as along the continental shelf break and near prominent underwater features, including seamounts, ridges, and troughs (see Fig. A2b). For example, the Weddell Sea, Ross Sea, and Prydz Bay regions display strong CPE values, underscoring the critical role of underwater topography in modulating the occurrence of sea-ice leads as discussed in Reiser et al. (2019) for the Antarctic and in Willmes et al. (2023) for the Arctic.

## 4  Discussion

The analysis of monthly LF anomalies from 2003 to 2023 reveals distinct temporal and spatial variations across the Southern Ocean. Figure 6 indicates that certain regions and time periods exhibit significant deviations in LF from the climatological mean. These anomalies are superimposed to the general seasonal patterns of leads (Fig. 4) and can be used to get a better insight into the large-scale drivers of sea-ice break-up and lead dynamics (Kimura and Wakatsuchi, 2011). We hypothesize that the presented monthly anomalies in regional lead frequencies are driven by a combination of large-scale atmospheric and oceanic processes. However, the relative importance of those two components is yet to be estimated. In this study, we therefore present first insights into this new lead dataset to describe the spatial and temporal variability of the Southern Ocean lead dynamics.

Our findings support the argument that sea-ice leads are widely distributed across the Southern Ocean, with particularly high frequencies of occurrence along the coastline, over the continental shelf break and ridges (see Fig. 2a), which underlines the findings of Reiser et al. (2019) and extends upon their results. In many of these regions, especially near the coast, these patterns are caused by flaw-leads and polynyas, which are opening mainly due to wind-driven processes and thereby promote new ice formation and the production of cold and dense shelf water, a precursor to Antarctic Bottom Water, which represents a key element of the global climate system (Goosse and Fichefet, 1999; Jacobs and Weiss, 1998; Silvano, 2018). Therefore, knowledge about the seasonal, inter-annual and spatial variability of lead dynamics and their drivers is one of the crucial fields to put forward our understanding of global ocean circulation and climate (Bindoff, 2000; Jacobs, 2004). The presented dataset does not include the presence of polynyas, which were large enough to cause a signal passive microwave data. However, it can be very well combined with long-term polynya observations as given for example in Lin et al. (2024).

The occurrence and variability of polynyas in the Southern Ocean and the consequent new ice production and salt release were investigated in many previous studies (e.g., Golledge et al., 2025; Macdonald et al., 2023; Tamura et al., 2008; Tamura et al., 2016). These investigations did, however, mostly not take small-scale leads within the pack ice into account, which is mainly because satellite data were used that did not allow to resolve sea ice surface features at the km scale or below. The data we present here show that leads are ubiquitous in the Southern Ocean pack ice during wintertime. This supports the need for more detailed information about sea-ice dynamics within the pack ice to improve our understanding of ocean-sea ice-atmosphere interactions and climate prediction in the future (Mohrmann et al., 2021).

The continental slope plays a key role in shaping spatial lead patterns. One of the possible explanations for this feature is ice divergence in these regions, driven by the ASC and tidal flows (Stewart et al., 2019; Heil et al., 2008; Hutchings et al., 2012). Also, regions such as Maud Rise (Beckmann et al., 2001) or Gunnerus Ridge exhibit enhanced tidal currents, which can increase



sea-ice velocity and divergence. In addition to tidal flows, the thermal variability at Maud Rise is significantly influenced by
advection and mesoscale eddies (Gülk et al., 2023). These processes underscore the interplay between bathymetry, ocean
circulation, and lead formation in this region. Our observations provide evidence that this also facilitates lead formation in
these regions and over seafloor ridges in general. The opening of leads during winter triggers strong energy exchange processes
that can alter both, the stratification of the upper ocean (Venables and Meredith, 2014; Cohanim, 2021) and the atmospheric
boundary layer (Heinemann et al., 2022; Marcq and Weiss, 2012). In this context, the role of leads in shaping the feedback
between sea ice, atmosphere and ocean must not be neglected and requires a robust identification of forcings, for which we
think the presented lead climatology will yield a useful basis.

In light of the observed trends (e.g., Schroeter, 2023; Eayrs et al., 2021) and recent changes in Antarctic sea ice (Suryawan-
shi et al., 2023; Turner et al., 2016), a detailed look at what happened within the pack ice becomes increasingly important
and makes sea-ice lead dynamics get growing attention. Our results indicate weak, but significant trends in wintertime lead
frequencies in the period between 2003 and 2023, which are mostly positive (Fig. 5 and Table 1). We do not find a signal in the
presented lead climatology that can be associated with the observed negative sea-ice extent anomalies after 2015 (Zhang et al.,
2022). Likewise, the observed minimum Antarctic sea-ice extent in July 2023 (Ionita, 2024) is not accompanied by more leads
in the remaining pack ice. This points out that sea-ice compactness during winter, at least south of the MIZ, seems not to be
affected by the processes responsible for the overall sea-ice decline in the Southern Ocean since 2016. Although the detected
LF changes in the 21-year period are small, Table 1 exhibits some spatial and seasonal contrasts in the sign of the slope, which
might point to a weak, but spatially distinguishable contrast in lead changes over the last 21 years. The LF trends presented
here exclude the contribution of low-quality data (LFA < pIE) to provide high confidence for the observed change signal. If
the significant increase in LF in certain regions — linked to larger areas of low-quality data after 2020 — is indeed real, then
substantial changes in lead dynamics may have occurred since. For this to be confirmed, however, a more in-depth analysis
with additional datasets would be necessary. LF after 2020 tends to be larger compared to previous years (Fig. 6a) in almost all
of the observed regions. This observation coincides, however, with higher pIE values and the associated decrease in confidence
of monthly LF values, due to an increase of data gaps. While a direct analysis of detailed cloud cover changes is beyond the
scope of this paper, it's important to acknowledge the impact of clouds on lead detection using thermal-infrared satellite data.
E.g., Goessling et al. (2025) examine the trend in ERA5 low cloud cover from 2013 to 2022 and identify an increasing coverage
of low clouds in the Weddell Sea in the last couple of years. Detected changes in monthly LF may also partly be attributed
to changes in shelf ice edges and associated local modifications in the presence of open water areas that have occurred since
2002. The lead maps used in this study are based on the land mask that is provided with the Moderate Resolution Imaging
Spectroradiometer (MODIS) ice surface temperature product (see Reiser et al., 2020).

For the pan-Antarctic sea-ice extent as a broader quantity, the connection to atmospheric and oceanic drivers were demon-
strated in a wide range of published research papers. E.g., changes in atmospheric circulation patterns, such as the Southern
Annular Mode (SAM) and the El Niño-Southern Oscillation (ENSO), significantly influence the extent of Antarctic sea ice as
well as the position of the MIZ (Yuan, 2004; Hall and Visbeck, 2002; Maksym, 2012). These modes affect wind strength and
direction, which in turn impact sea-ice dynamics and presumably also the opening/closing of leads. In addition, sea surface



temperature (SST) anomalies in subtropical regions, such as the Indian Ocean Dipole (IOD), show a clear impact on the com-
pactness, drift patterns and trends of sea ice in the Southern Ocean (Blanchard-Wrigglesworth et al., 2021). SST anomalies
propagate through atmospheric circulation patterns, which influence sea ice transport and temperature advection (Yu, 2023).
As far as trends are concerned, the atmosphere is generally considered the primary driver, while the ocean is important for the
seasonality of the trend patterns (Holland and Kwok, 2012; Hobbs et al., 2016). However, generally, previous investigations of
large-scale modes and their impact on sea ice are mostly constrained to sea-ice extent or low-resolution sea ice concentration.
Therefore, the dataset we present here offers a new opportunity and supplement to existing datasets to conduct thorough inves-
tigations on the relationship between high-resolution sea-ice compactness in the Southern Ocean and its large-scale drivers.

## 5 Conclusions

In this study, we present a new dataset on the spatial distribution and temporal variability of wintertime sea-ice leads in the
Southern Ocean for the months of April to September for the 21-year period from 2003 to 2023. The data include monthly
lead frequencies and their potential integration error on a per-pixel basis at 1 km$^2$ spatial resolution. Monthly and annual mean
lead frequencies, along with their anomalies and trends are derived to provide first insights into long-term lead dynamics.
The results reveal pronounced spatial patterns in the long-term sea-ice lead climatology with higher lead frequencies observed
along the Antarctic coastline, particularly in the Weddell and Ross Seas, over continental shelf breaks and bathymetric features,
such as ridges. This provides strong evidence for a significant role of ocean currents in shaping lead dynamics in the Southern
Ocean. On the monthly scale, we can identify a pronounced, but regionally different, seasonal cycle in lead dynamics and
the presence of strong monthly anomalies in different regions across the Southern Ocean. We suggest these anomalies to be
potentially attributed to atmospheric forcing. We find small but significant long-term trends in lead frequencies in the Southern
Ocean with regionally differing signs of change. The fact that no strong trend is found indicates relative stability in wintertime
sea-ice compactness in areas south of the MIZ, while the overall sea-ice extent in the Antarctic has shown substantial variability
and recent declines, particularly post-2016. This work provides initial results for future investigations into the complex pan-
Antarctic interplay between sea ice, ocean and atmosphere. The presented data can help to better estimate and untangle the
individual contribution of atmospheric and oceanic forcings on sea-ice lead dynamics in the Southern Ocean.





# Appendix A

## A1   LF anomaly variability with respect to dgF

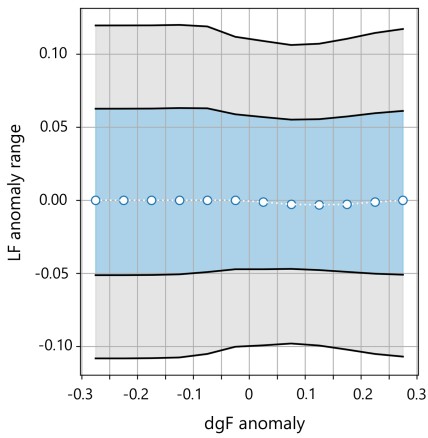

**Figure A1.** Range of LF anomalies with respect to corrected monthly mean LF as a function of the coincident dgF anomaly: Median (circles), 1 (blue) and 2 (grey) standard deviations.

Figure A1 presents the distribution of LF anomalies as a function of the coincident dgF anomaly class. The figure highlights the independence of the LF variability from dgF anomalies.

## A2   ORAS5 current velocity and ocean depth

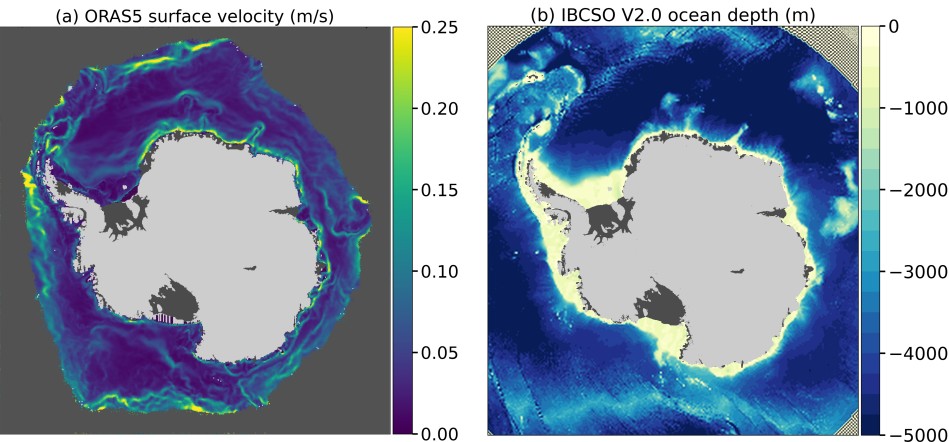

**Figure A2. (a)** Overall mean of ORAS5 surface current velocity for 2003–2023 (April to September) and **(b)** IBCSO2 ocean depth.



Figure A2a represents the mean surface current velocity for the Southern Ocean during the winter months (April–September), 2003–2022. The highest current velocities are observed near the Antarctic Circumpolar Current and along the continental

shelf. Figure A2b shows bathymetry data from the IBCSO version 2 dataset. The alignment of high LF occurrences with these features emphasizes the critical role of underwater topography in shaping spatial LF patterns (see section 3.8).

*Data availability.* Monthly lead frequencies (LF) and potential Integration Error (pIE) data for the period April 2003 to September 2023 are currently under consideration for publication in NetCDF format on PANGAEA (AntLeads: Monthly wintertime sea-ice lead maps for the Antarctic, April–September, 2003–2023, DOI is pending and will be included in the final version). ORAS5 data (Zuo et al., 2019) were

downloaded from the Copernicus Climate Data Store (https://cds.climate.copernicus.eu/), and version 2.0 of the International Bathymetric Chart of the Southern Ocean (IBCSO, Dorschel et al., 2022) was acquired from the General Bathymetric Chart of the Ocean (GEBCO) gridded bathymetric datasets (https://gebco.net).

*Author contributions.* UD and SW processed and analysed the data and drafted the main script. The final version was prepared with contributions from all co-authors including GH. All authors have read and agreed to the submitted version of the manuscript.

*Competing interests.* The authors declare that they have no conflict of interest.

*Acknowledgements.* The research was funded by the Deutsche Forschungsgemeinschaft (DFG) in the framework of the priority program "Antarctic Research with comparative investigations in Arctic ice areas" (SPP1158) under grant WI 3314/6-1. All processing was done in Python3. The authors want to thank the ECMWF and GEBCO for providing the data used in this study. The publication was funded by the Open Access Fund of Universität Trier and the German Research Foundation (DFG) within the Open Access Publishing funding program.



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
