# Peer review of "Southern Ocean sea-ice leads: First insights into regional lead patterns, seasonality and trends, 2003–2023"

_EGUsphere, 2025_

## Author Comment (AC1)

**Response to Referee 1 Comments**

We show referee comments in black text, our response in blue and changes inserted to the manuscript are put in *italics*.

A review of "A new dataset of Southern Ocean sea-ice leads: First insights into regional lead patterns, seasonality and trends, 2003–2023" by Dubey and coauthors.

This is a well-written, detailed and generally well-supported characterization of leads in Antarctic sea ice, and is an aggregation and subsequent quantitative analysis of the underlying dataset of daily leads from Reiser et al, 2020. I find that the paper is well-structured, and that the content fits well within The Cryosphere (as opposed to ESSD). Furthermore, the content is of great interest to those of us trying to disentangle the rollercoaster ride of Antarctic sea ice over the past decade, and places many other works in new context.

We appreciate the time and effort you have taken to review our manuscript. Your constructive comments and suggestions have been very valuable in improving the quality and clarity of our work. We have carefully addressed each of your points.

I have only one major comment, detailed below, regarding Section 3.6. All others can be regarded as quite minor, and I think the author team will deal with these without too much trouble.

Major:

The whole of Section 3.6 is a little perplexing. If my understanding is correct, based on the description presented, the following procedure is followed to determine regional trend magnitude/significance:

Thanks a lot for pointing out issues with the perspicuity of this part of the analysis.

1) Each grid cell, in each month, has 21 data points (one point per month over the 21 year dataset).

Correct.

2) In each grid cell, the linear trend is calculated from these 21 points.

Correct.

3) For each grid cell, the trend is either determined to be "significant" or "insignificant" - based on what criteria? A "5% level" is mentioned but this isn't really sufficient information to replicate your method. Is a linear regression calculated How many degrees of freedom used? Two tailed or one tailed? Etc.

A two-tailed linear regression is used with all 21 data points, meaning that the trend is significant if the probability that the given slope deviation from zero occurs only by chance is smaller than 5%. We've added the missing information to Section 3.6.

4) Then these grid cells are aggregated into regions - but only significant (positive or negative) pixels are aggregated? It's a little baffling why perfectly valid (but insignificant) pixels are left out of this calculation.

Thanks a lot for this remark. We agree that the description of the trend detection needs refinement and that regional averaging using only significant grid points is a misleading approach to highlight regional differences in long-term LF changes. We now chose instead to show a map of the overall LF change based on annual LF values to better point out regional characteristics of the overall LF change in the 21-years period.

5) Then in Table 1, it's not clear what a value of "-" means. Is this 0? Or are you taking 0 to also mean insignificant, and also reporting that as "-"? In that case, it's very suspicious that a value of 0.1 just happens to coincide with your threshold of significance.

Basically I'm not convinced that leaving out insignificant pixels is the right thing to do, and not convinced that all values in Table 1 are significant trends.

Again, thanks for this remark. We have now replaced the trend table with a map (new Figure 7) to avoid confusion about regionally averaged trend values.

Minor, referenced by line number where possible:

Abstract/Intro:

- recommend change "can be inferred" to "are shown"

Changed.

- lots of "rather" in this sentence

We have reduced the use of "rather" in the sentence: *Rather small changes in lead occurrence over the 21 years suggest stable wintertime sea-ice compactness despite the observed strong fluctuations and recent anomalies in sea-ice extent.*

- I think Reiser et al 2020 could probably claim "first results on the spatial...." - reword to make the claim accurate.

We have reworded: *Expanding upon previous work of lead detection in Antarctic sea ice, this study provides first results on the long-term regional, seasonal, and inter-annual variability of sea-ice leads in the Southern Ocean and can thereby contribute to an improved understanding of air-sea ice-ocean interactions in the climate system.*

- an inconsistency in the use of a hyphen between "sea" and "ice" where it precedes a noun. E.g. if you use "sea-ice leads", as you do, then you should use "sea-ice drift" and "sea-ice deformation". This is inconsistent in many places.

We have ensured consistent use of the hyphenation "sea-ice" when preceding a noun."

- regarding the citations for ice-albedo feedback, I recommend to cite an Antarctic-specific example - perhaps https://doi.org/10.1029/2005JC003447

Thanks for your recommendation. We have included the suggested study (https://doi.org/10.1029/2005JC003447).

- the use of "winter" is a little too unspecific in many places of this manuscript. April isn't exactly a winter month. Also line 108, 123, etc...

In our study, we have considered the wintertime/winter months as April to September. We have specifically added "April to September (hereafter referred to as winter)" to point this out clearly in the revised version (see line 44).

- deduct isn't quite the right word - is "deduce" better?

We have replaced "deduct" with "deduce".

- this sentence should make clear that it's an aggregation of the Reiser dataset, and also give the months in which the technique is valid.

Thank you for your feedback, we have revised: See response to next comment.

Introduction in general - it's not actually clear, especially for someone unfamiliar with the Reiser dataset, what you're doing. I think the introduction would benefit from more description of the Reiser dataset. E.g. what instrument? Aqua only? Or Terra too? What are the drawbacks of just using the Reiser dataset (without the aggregation that you present here)?

We have now changed this paragraph to:

*The potential of thermal-infrared satellite data for the detection of sea-ice leads has been demonstrated for the Southern Ocean in a previous study (Reiser et al., 2020). Their lead detection relies on identifying significant positive ice surface temperature anomalies from the Moderate Resolution Imaging Spectroradiometer (MODIS) onboard NASA's Aqua and Terra satellites and the associated ice surface temperature data set (MOD29/MYD29, Hall and Riggs, 2015), which includes the MODIS cloud mask to flag pixels affected by cloud cover. The method of Reiser et al. (2020) thereby detects the presence of leads in both hemispheres during winter months (here: April to September) and also identifies false lead detections due to surface temperature anomalies that result from deficiencies in the MODIS cloud mask. As a result, leads can be detected on a daily basis at a spatial resolution of 1 km², however, due to high cloud fractions in the Antarctic as compared to the Arctic, the daily lead maps in the Southern Ocean suffer from extended data gaps, that need to be dealt with when it comes to investigating the long-term regional lead patterns, seasonality and trends. The primary objectives of this study are therefore to deduce a continuous and gap-free monthly lead dataset from the daily lead data obtained from Reiser et al. (2020) and to use this dataset to infer the spatial distribution of sea-ice leads, quantify temporal trends and compare regional differences for the months of April to September in the 21-year period from 2003 to 2023.*

Data and methods:

49-50 - are these anomalies in space, time or both?

It is spatial surface temperature anomalies, not temporal. We've pointed this out in the revised version.

- this is the first mention of the April - Sept data validity - I think it needs to be done in the intro.

This is explicitly stated now in line 44.

- please give some ideas of other datasets to combine.

We've added a reference for polynya data.

- "annual" is only April - September, right?

Yes, that's right. We've changed to "annual (winter)".

- I think "averaging" is more correct than "integrating"

Changed to "averaging".

- there may be nothing that can be done within the style guide, but that subscript r is so small!

Subscript r is bigger now.

- You take anomalies from the long-term mean? Does this mean all months from April to September as your baseline? Wouldn't it make more sense to just choose the same month as your baseline?

Yes, the anomalies are derived based on the long-term winter mean. We understand your argument, however, we think that using the winter mean as the baseline allows for a more robust and conservative correction, because otherwise a seasonal dependence of the correction could be introduced.

- sentence beginning "This approach" is hard to interpret. Do you mean "not to avoid..."?

We changed to: *This approach seeks to prevent over-interpretation of large deviations from the long-term mean, while enhancing confidence in the observed signals, i.e., LF anomalies.*

- This section would benefit from first stating your motivation for looking at surface currents. E.g. "Surface currents are the main determinant of leads, so...."

We have changed to: *For a preliminary evaluation of confidence in the identified lead patterns and to assess the influence of ocean dynamics on lead formation, we use ocean surface current data, provided by the European Centre for Medium-Range Weather Forecasts (ECMWF) Ocean Reanalysis System 5 (ORAS5) dataset (Zuo et al., 2019). Surface currents can play a crucial role in modulating lead formation by influencing sea-ice drift, mechanical stress and ice deformation.*

- This CPE technique isn't quite straightforward, and also no reference is provided.

We have added: *The CPE represents a simple measure of spatial coincidence by comparing the exceedance probability of two variables in a spatial field. This approach is inspired by the Willmes et al. (2023), where it was used to analyze spatial patterns of leads in the Arctic Ocean.*

- "study is" to "study are" (data is a plural)

Changed to "study are".

- DBM is never used again, so don't define the acronym

Done.

Results

- "most dominant" is subjective - reword

We have reworded: *The most prominent spatial lead patterns are found in the Weddell and Ross Seas along the shelf break with values exceeding 0.25, which means that leads were present on more than 25 % of days during winter.*

- a publication showing the fast-ice long-term mean distribution would be important to reference here - https://doi.org/10.5194/tc-15-5061-2021

Thanks for your remark! You are absolutely right, we know this publication and acknowledge that it must be referenced when it comes to describing fast ice in the Southern Ocean. We definitely want and need to include this reference in a revised version. Thank you for pointing us towards this deficit.

Fraser et al. (2021) is included now!

- MIZ is defined here but would be better to define 2 lines earlier.

MIZ has been defined two lines earlier.

- this sentence structure (sentence beginning "Areas, ..") sounds unnatural. This structure is used in other places too (e.g., figure captions).

Changed to: Areas that do not meet the LFA > pIE criterion is highlighted in green, indicating low-quality data.

and Fig 4 - Somehow it's a little strange to me for this figure to be presented as an anomaly. First of all, what's it an anomaly from? I guess Fig 2a, right? Secondly, I kind of prefer to see this figure presented as a mean for each month, rather than anomaly from the mean. Without seeing the means, it occurred to me that the authors maybe already explored this and decided the anomalies are more meaningful or easy to interpret. That's fine, if that's the case, but for baseline characterisation I find that means can sometimes be more important. Just a thought. I have no problem with the maps in Fig 3, and somehow in my mind it's clear that Fig 3 should be presented as anomalies, but somehow it doesn't seem as natural for the baseline monthly figures in fig 4.

Thank you for your feedback. You are correct that Fig 4 presents the mean monthly LF anomalies relative to the overall mean LF. The decision to present the data as anomalies is motivated by our goal to highlight mean seasonal variations in LF patterns, which we believe are more illustrative when given as seasonal anomalies.

To address your concern, we here present the mean LF for each month as suggested by you. It would also be okay for us to use the mean monthly LF as shown here in Figure 4 instead, but we actually would prefer sticking to the anomaly version.

[Figure]

- unnatural sentence around "months, where"

Changed to "months in which".

Fig 5 caption - can you state the colours used for months in the caption? It's a bit cramped and I think these would help.

We've now stated: *Figure 5. Inter-annual variability of monthly mean lead frequency (April to September) in the Southern Ocean and its sub-regions from 2003 to 2023. Sub-regions are identified in the inset map. Individual months are color-coded as follows: April (blue), May (green), June (red), July (orange), August (brown), and September (black). Month dots without color fill indicate low-quality (LFA < pIE) data point coverage above 50%. Note the different scales on the y-axes for monthly LF.*

- Fraser et al 2009 (https://doi.org/10.1109/TGRS.2009.2019726) showed that leads often coincide with an erroneous "cloud" determination from the MODIS cloud mask (see their Fig 6). Is this perhaps what you're seeing here? Does the Reiser dataset rely on the MODIS cloud mask?

Yes, the dataset relies on the MODIS cloud mask. A false classification of leads due to deficits in the cloud mask is taken care of in the Reiser dataset by introducing an artefact class in the lead detection. Pixels assigned to this artefact class are being interpreted as a data gap in this study (adding up to dgF). The higher uncertainty after 2020, however, results from higher cloudiness due to the MODIS cloud mask (larger data gaps).

- a good idea to reference the map of long-term mean fast-ice persistence from Fraser et al., 2021 (https://doi.org/10.5194/tc-15-5061-2021)

Absolutely, we are sorry for not having referenced this important publication in this context yet. Thanks for your remark! We have referenced Fraser et al. (2021) for long-term fast-ice persistence (https://doi.org/10.5194/tc-15-5061-2021).

- "inverted fast-ice frequencies" - I see what you're saying but this might be overstating it. E.g. your plot in Fig 8b, right column shows the climatology of lead frequency increasing throughout the winter, but fast ice also increases throughout the winter in this region.

The statement "inverted fast-ice frequencies" actually only refers to the LF climatology presented in Figure 7 (now Figure 8 in revised version of the manuscript), not to the temporal evolution.

We've changed it to: *The LF values in these regions generally correspond to areas of frequent fast-ice occurrence (see Fraser et al., 2021), with low LF often indicating the presence of stable fast ice.*

- missing an accent in "Adelie". NB also typo in Fig 7, box 4 "Adleie".

Corrected the spelling of "Adélie" and fixed the typo in Figure 7 (now Figure 8 in revised version of the manuscript), box 4.

- I feel like Maud Rise is commonly-enough known to stand without a lat/lon, but Gunnerus Ridge probably isn't.

We have now indicated the Maud Rise and Gunnerus Ridge with arrows in Figure 10 (now Figure 11 in revised version of the manuscript).

- "These anomalies are superimposed..." not really - Fig 4 is *also* anomalies, not "general seasonal patterns".

We've reworded: *These anomalies complement the seasonal variability of monthly mean lead frequency anomalies shown in Fig. 4 and can be used to get a better insight into the large-scale drivers of sea-ice break-up and lead dynamics (Kimura and Wakatsuchi, 2011).*

---

## Author Comment (AC2)

We show referee comments in black text, our response in blue and changes inserted to the manuscript are put in *italics*.

Leads is an important indicator of Antarctic sea ice deformation, and their changes reflect the vulnerability of Antarctic sea ice. However, it is a pity that the relevant research has not attracted enough attention, partly because such a small-scale phenomenon is difficult to detect or simulate. This paper using the latest high-resolution leads detection data push the research in this field to make significant progress, making me exciting. For the first time, it reported regional and seasonal trends in the Antarctic leads. I think it would be important circumstantial evidence that Antarctic sea ice is weakening under the dramatic climate change of the last two decades. I also sincerely commend the authors for their willingness to share leads dataset.

We thank the reviewer for insightful comments on our manuscript. Your comments are valuable and will help improve the quality of the study. We have carefully addressed each of your points.

I believe the conclusion of the article is attractive enough, but a few changes are needed.

1. This paper claims to have developed a new dataset, but the improvement over Reiser's work is very limited: it seems to have only filled in missing values with multi-year means and reduced the resolution to monthly. Therefore, I think it is overstated to emphasize this point in the title and abstract.

   Thank you for your feedback. We disagree, however, on this point. While our dataset is based on the method from Reiser et al. (2020), our main contribution is a detailed analysis of the spatial and inter-annual variability of sea-ice leads. Our study builds on Reiser's work by examining how lead occurrence in Antarctic sea ice changes over the last two decades, across regions, and between seasons and years. Reiser et al. (2020) focused on developing a methodology for detecting leads from thermal infrared satellite imagery and presented a general climatology based on 17 years of data. In contrast, our study uses an extended 21-years gap-corrected dataset and presents a detailed analysis of seasonal patterns, regional differences, and year-to-year variability – topics that were not addressed in earlier study. Therefore, we believe it is appropriate to emphasize the dataset, as it forms the basis for the new analyses and insights presented. The consistently processed monthly dataset enables exploration of spatial and temporal patterns not covered in previous work. We therefore consider the current title and abstract to accurately reflect the study's scope and contributions.

2. Figure 6 shows that there are many misses after 2020, but this period contributes to most of the increasing trend of East Antarctica leads, which weakens the reliability of the trend and would shake the foundation of this article. It is necessary to check the code and recalculate the leads frequency during this time. There are missing AMSR-based sea ice concentrations in 2011-2012, and the authors do not explain how they dealt with this period.

   Thanks for this remark. The identified increase in leads is not a coding artefact, but results from larger data gaps in this period as mentioned in the paper. The increase in missing values after 2020 in the East Antarctic in Figure 6 reflects a decrease in the number of cloud-free days available for reliable lead detection using thermal infrared imagery. This is because the dataset depends on the MODIS cloud mask, and increased cloudiness after 2020 has resulted in larger data gap frequency (dgF). We also do not consider the

observed increase in leads a foundation of this article because we explicitly discuss that this observation cannot be attributed to a significant trend due to the mentioned decrease in data quality.

Regarding the AMSR-based sea-ice concentration data gaps during October 2011 to June 2012, we agree that additional explanation is required. We used AMSR-E/AMSR2 data only to define a mask for the lead analysis by sea-ice extent (sea-ice concentration >=15%). For the months of April to June 2012, the mean sea-ice concentration as derived from the period with available AMSR-E/AMSR2 data was used. This information was missing in the text so far and will be added in the revised version. Thanks again for your comment.

We have added: *For the months of April to June 2012, due to the gap in AMSR-E and AMSR2 data between October 2011 and June 2012, the mean sea-ice concentration as derived from the period with available AMSR data is used.*

Some small comments and presentation issues are also worth noting.

1. Section 3 needs to be better organized, especially Section 3.5 and the following paragraphs. Analysis that is redundant and jumpy with the previous description is confusing. The authors need to find a clear logical thread to rearrange how to present these results.

   We are actually not sure how to comprehend this remark in detail. According to the comments provided by Reviewer 1 we have applied changes in Section 3 in the revised manuscript with respect to the trend analysis and also improved some remarks and associated conclusions to better point out the main findings.

2. Line 80: What's the meaning of "absolute" pIE?

   "Absolute" here means that the pIE has the same unit as LF and is not to be interpreted as a relative error.

3. Line 126: For color blindness, please don't put red and green together. You can replace green with cyan. The same goes for the other figure.

   We appreciate your suggestion. To ensure accessibility for individuals with color blindness, we have replaced green with yellow instead of cyan for the revised version. We believe this will improve the clarity and visibility of the figure for all readers.

4. Line 160: Is the trend significant? It seems that many of the trends in this paper lack p-values.

   In line 160 we only refer to qualitative changes in LF that are visible in the presented time series and do not want to state a presence of trends. That is why this statement is followed by the sentence: "It has to be noted, however, that a large part of this observed increase is attributed to an increase in low-quality data points during the last 3 years of the observed period." We suggest to add the remark: *"which does not allow to infer a trend in LF."*
   In section 3.6 we specifically address trends, associated p-values and limitations for potential conclusions. This section has also been worked over in the revised version of our manuscript to better point out the mentioned issues.

5. Line 211: Please explain "pronounced regional differences" in detail and explain how leads and fast ice are physically related.

   Thank you for your comment. By "pronounced regional differences", we refer to the distinct patterns of LF observed in different regions of the East and West Antarctic. These

differences are particularly evident when comparing the coastal regions with the inner ice pack. The presence of fast ice can locally influence wind and ocean patterns, affecting where leads are likely to form and areas with more frequent flaw-leads generally have less stable fast ice.

We added the following sentence in the revised version: *"The presence of leads can affect the dynamics of fast ice by altering local wind and ocean currents, which can lead to changes in ice thickness and extent and fast ice can also influence the formation of leads by providing a barrier that modifies the stress distribution in the surrounding ice cover, potentially leading to fractures and the opening of leads (Kim et al., 2018, doi: https://doi.org/10.1017/S0954102017000578 )."*

6. Line 233: Given the large uncertainties in ocean current simulations, I recommend that you validate your results using multiple analyses/reanalysis.

Thanks for your recommendation. There is definitely potential to compare spatial lead patterns with different ocean analyses, but we think this would reach beyond the scope of this paper and be misplaced in the presented context. We will consider incorporating additional analysis/reanalysis in future work.

7. Figure 7. This figure does not seem to have any difference from Figure 2a. This one looks a bit prettier, so you can keep just this one.

We would prefer to stick to both figures as they both provide a different focus on spatial patterns in the lead climatology. While Figure 2a presents, (a) an extension of the climatology presented by Reiser et al. (2020) and (b) it also better points out lead patterns in the inner ice pack and their relation to bathymetry. Figure 7 (now Figure 8 in revised version of the manuscript), in contrast, highlights the spatial lead patterns in coastal areas and how these are connected to fast-ice areas in some regions.